# Experimental Estimation of Deviation Frequency within the Spectrum of Scintillations of the Carrier Phase of GNSS Signals

Vladislav Demyanov [1,*], Ekaterina Danilchuk [2], Yury Yasyukevich [1] and Maria Sergeeva [3,4]

1 Institute of Solar Terrestrial Physics, Siberian Branch, Russian Academy of Sciences, 664033 Irkutsk, Russia; yasukevich@iszf.irk.ru
2 Department of Physics, Irkutsk State University, 664003 Irkutsk, Russia; danilchuk.k@mail.ru
3 SCiESMEX, LANCE, Instituto de Geofisica, Unidad Michoacan, Universidad Nacional Autonoma de Mexico, Antigua Carretera a Patzcuaro 8701, Morelia 58089, Mexico; maria.a.sergeeva@gmail.com
4 CONACYT, Instituto de Geofisica, Unidad Michoacan, Universidad Nacional Autonoma de Mexico, Antigua Carretera a Patzcuaro 8701, Morelia 58089, Mexico
* Correspondence: vv.emyanov@gmail.com; Tel.: +7-950-051-3095

**Abstract:** The term deviation frequency (fd) denotes the boundary between the variable part of the amplitude and phase scintillation spectrum and the part of uninformative noises. We suggested the concept of the "characteristic deviation frequency" during the observation period. The characteristic deviation frequency is defined as the most probable value of the deviation frequency under current local conditions. Our case study involved GPS, GLONASS, Galileo and SBAS data under quiet and weakly disturbed geomagnetic conditions (geomagnetic storm on 16 April 2021, $Kp_{max} = 5$, $SYM-H_{min} = -57$ nT) at the mid-latitude GNSS station. Our results demonstrated that the deviation frequency for all signal components of GPS, GLONASS and Galileo varies within 15–22 Hz. The characteristic deviation frequency was 20 Hz for the mentioned GNSS signals. The SBAS differs from other systems: deviation frequency varies within 13–20 Hz. The characteristic deviation frequency is lower and equal to 18 Hz. We suggest the characteristic deviation frequency to determine the optimal sampling rate of the GNSS carrier phase data for the ionospheric studies. In turn, the deviation frequency can be considered as a promising index to estimate the boundary of non-variability of the ionosphere.

**Keywords:** ionosphere; scintillations; carrier phase; GNSS; GPS; GLONASS; Galileo; SBAS; GNSS signals; deviation frequency

## 1. Introduction

Global Navigation Satellite Systems (GNSS) form part of a technological basis for different applications [1]. Their data is widely used for fundamental research tasks in different fields, for example in geodynamics [2], radio propagation environment including the GNSS remote sensing (GNSS-RO) [3] and GNSS Reflectometry of Earth surface (GNSS-R) [4]. In particular, one of the important geophysical studies that was carried out on the basis of GNSS signal processing was the study of the Earth's ionosphere and upper atmosphere [5–7] and their impact on different levels of applications [8–12].

In many cases the main measured parameter of satellite vehicle (SV) signals is the carrier phase that is characterized by the lowest multipath noise level and the highest measurement accuracy. Various stochastic techniques report normally distributed carrier phase noise of 2 mm and code pseudo range noise of 0.5–0.8 m [13]. Such precise measurements allow us to detect the effects of rather weak geophysical events and eventually reconstruct the structure of the ionosphere.

Determination of optimum sensitivity of the carrier phase lock loop (PLL) is an important task for the remote sensing of the ionosphere. The PLL sensitivity depends on both the internal noises and the sampling rate of carrier phase measurements [14]. The PLL

sensitivity is considered as an optimal if the probability of detection of a weak event at the background of the uninformative noise within the phase variation spectrum is the highest at a given noise level and a given sampling rate.

On the other hand, the higher boundary in the spectrum of the phase variations (at which weak disturbances can still be detected) may be considered as the boundary of the regular ionosphere. Studies that are focused on estimation of this boundary and its dependence on observation conditions are of particular importance for geophysical research advances.

McCaffrey and Jayachandran [15] suggested the "deviation frequency" term to denote the boundary between the variable part of the amplitude and phase scintillation spectrum and the part of uninformative noise in the spectrum. Developing this idea, we suggest introducing the concept of the "characteristic deviation frequency" during the observation period. The characteristic deviation frequency is defined as the most probable value of the deviation frequency (fd) under current local conditions. We assume that the characteristic deviation frequency is an estimate of the optimal sensitivity of the PLL under the current conditions.

It is worth pointing out that the spectral slope of the phase variations is mostly defined with large-scale ionospheric structures such as manifestations of the acoustic-gravity waves in the form of large-scale travelling ionospheric disturbances. Such disturbances can be caused by auroral oval border pulsations (during geomagnetic storms and substorms), earthquakes of high magnitude, tsunamis, cyclones and other power processes in the lower atmosphere. Thus, the deviation frequency as a break point between the phase variations spectra (with the spectral slope < 0) and the noise part of the spectra (with the spectral slope ~ 0) varies depending on presence or absence of the large-scale ionospheric disturbances. This allows us to assume that the deviation frequency can be considered as a promising index to estimate the boundary of non-variability of the ionosphere.

The aim of this work is to estimate the characteristic deviation frequency under particular observation conditions. The research tasks are to reveal the features of variations of the characteristic deviation frequency for GPS, GLONASS, Galileo and SBAS satellites, types of satellite signals, time of day and the level of geomagnetic activity.

## 2. Data and Processing Method

The measurements of GNSS signal carrier phase were performed during 13 and 16 April 2021 with use of the multi-system multi-band navigation receiver Javad Delta-G3T connected to the RingAnt-G3T antenna [16]. The equipment was installed at the ISTP station (geographic coordinates 52.24°N, 104.26°E; geomagnetic coordinates 42.70°N, 177.43°E). The station belongs to the SibNet GNSS receiver network [17]. Figure 1 shows variations of Kp and SYM-H geomagnetic indices during the considered period.

The intensity of the geomagnetic storm on 16 April 2021 was weak. SYM-H reached its minimum value of −57 nT at 20:02 UT. The Kp index reached the value equal to 5 during the period of 18:00–21:00 UT. The main phase of the storm occurred in the local midnight sector. This means the minimal background electron concentration level. Such conditions imply the appearance of weak ionospheric disturbances registered at the border of the stationary of the ionosphere. Therefore, we chose the period of this storm for our analysis.

The level of carrier phase measurement noises significantly differs for signals of different systems and for different signal components [18–20]. This issue is important for estimation of the characteristic deviation frequency in the spectrum of phase variations and scintillations. Hence, the carrier phase noise for the signal components at L1, L2 and L5 frequencies of GPS, GLONASS, Galileo and SBAS satellites were studied first (Table 1). The measurements were performed with a 50 Hz sampling rate. The signal components description mentioned in Table 1 can be found in the data format description available at [21].

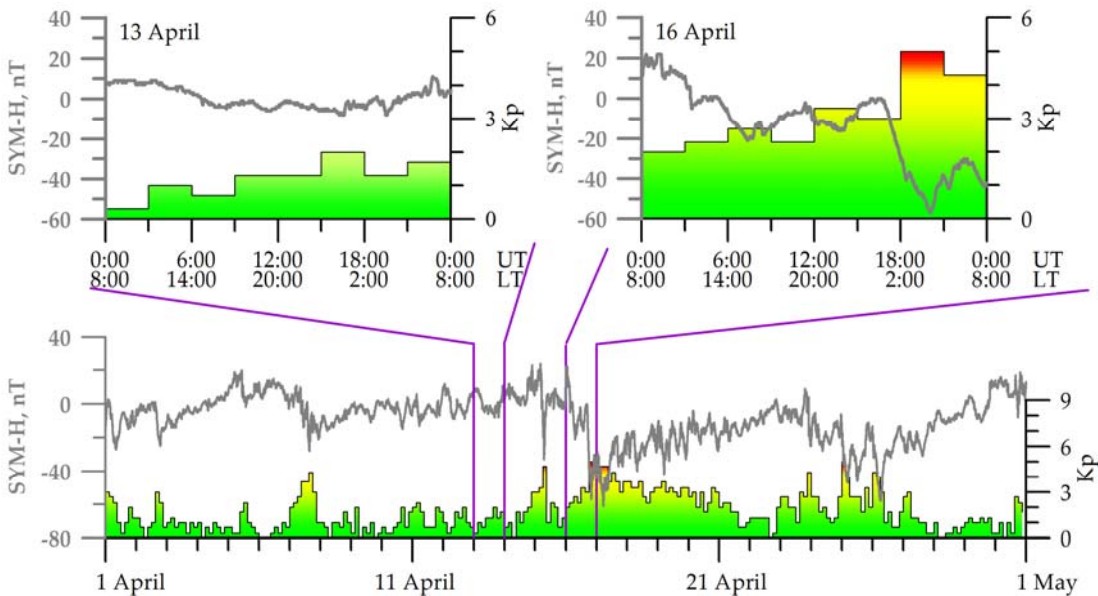

**Figure 1.** Variations of SYM-H and Kp indices during 13 April (**upper left** panel) and 16 April 2021 (**upper right** panel). General picture of SYM-H and Kp variations during 1–30 April 2021 (**lower** panel).

**Table 1.** Considered GNSS signal components.

| Navigation System | Signal Components | | | | | | |
|---|---|---|---|---|---|---|---|
| GPS | L1C | | L1W | | L2W | | L2X | L5X |
| GLONASS | | L1C | | L1P | | L2C | | L2P |
| Galileo | | | L1X | | | | L5X | |
| SBAS | | | L1C | | | | L5I | |

We applied the following procedure for deviation frequency estimation.

(1) The second order phase derivative was used to detect cycle slips and abnormal data in the carrier phase records. Cycle slip and anomalies in data yield sharp peaks of the phase derivative. All the carrier phase records that contain one or more events defined by Equation (1) were excluded from our analysis

$$dMax \leq d2L_j = \frac{L_{i+2} - 2L_{i+1} + L_0}{dt^2} \tag{1}$$

where *dMax* is the threshold defined empirically as *dMax = 10d2L$_{j-1}$*; *d2L$_{j-1}$* is the second order derivative of the carrier phase at the previous step of calculation; and *L$_i$* is the carrier phase at *i*-th time point (in cycles).

(2) Phase ambiguity was resolved by means of polynomial filtering. We used a 2-step algorithm to reduce the phase ambiguity. First, the main trend of the carrier phase was removed by means of linear interpolation as follows

$$\begin{aligned} IL_i &= a_0 + a_1 t_i \\ dL_i &= L_i - IL_i \end{aligned} \tag{2}$$

Further, the second-order trends of the carrier phase were removed by means of the 5-th order polynomial interpolation:

$$\begin{aligned} PL_i &= b_0 + b_1 t_i + b_2 t_i^2 + b_3 t_i^3 + b_4 t_i^4 + b_5 t_i^5 \\ LP_i &= dL_i - PL_i \end{aligned} \tag{3}$$

(3) The final de-trending of the carrier phase was performed by means of moving averaging window filter as follows

$$SL_{i+\frac{N}{2}} = \frac{1}{N+1} \sum_{k=0}^{N} LP_{i+k}$$
$$LS_i = LP_i - SL_i$$

(4)

The parameter of the filter $N$ defines the window width. Considering the results reported in [14], we chose the 5 min averaging window. According to [14], phase fluctuations caused by small-scale irregularities usually have a period of several seconds. Indeed, Pi et al. [7] reported a period of ~(2–13) s and Forte and Radicella [22] reported ~(0.4–5) s.

After the data preprocessing, a fast Fourier transform was applied to the de-trended phase data series $LS$ (4). We did not apply the Hanning window as in work [15], because of uncertainty of filter parameter choice under the unknown spread and behavior of deviation frequencies. The deviation frequency was determined at the logarithmic spectrum of phase variations as a "break point" at which the maximum decrease in the slope of the spectrum passes into near-zero decrease in a given frequency range. Figure 2 illustrates the example of deviation frequency identification from the logarithmic spectrum of phase variations for line-of-site to GPS satellite PRN07 on 13 April (panel a) and 16 April (panel b).

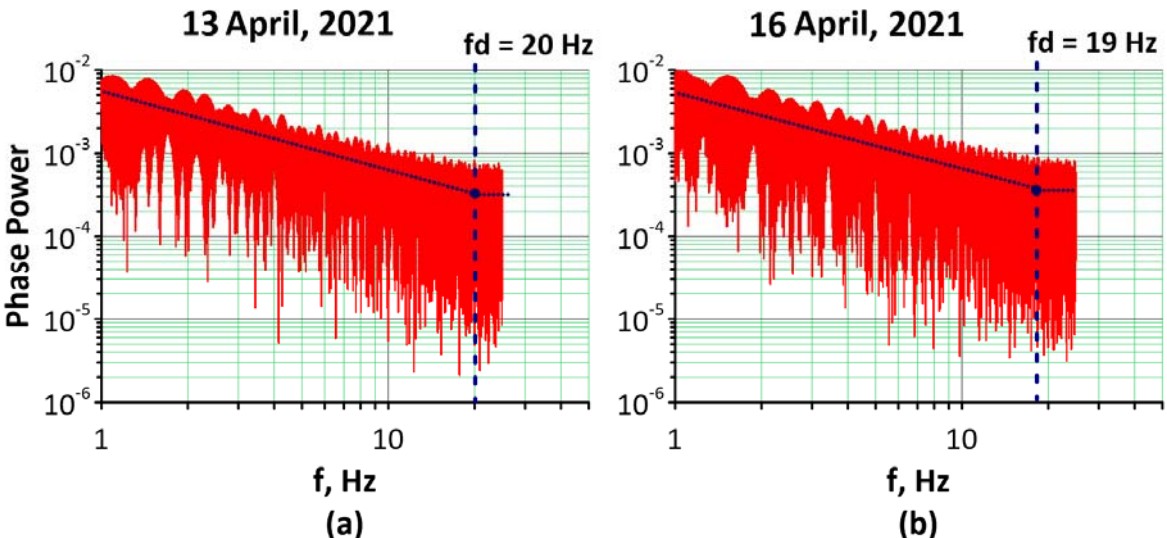

**Figure 2.** The identification of the deviation frequency under quiet (panel (**a**)) and geomagnetically disturbed (panel (**b**)) conditions.

When performing this analysis, the frequency range from 1 to 25 Hz in the spectrum was studied in which the deviation frequency was expected to vary. The "break point" defines the part of the phase variations spectrum where the expected power law character is shallowing towards white noise. Therefore, the carrier phase noise is a key issue to define the precise value of deviation frequency. That is why it is important to analyze the level of phase measurement noise for signals of different navigation systems, different frequencies and components.

The second-order derivative of the signal carrier phase was used as the noise magnitude (Please, see the methodology description in [14]). Table 2 provides root-mean-square (RMS) estimates for the noise of phase measurements for signals from different systems and components.

According to Table 2, the level of the carrier phase noise for the similar types of components and signal frequencies of GPS, GLONASS and Galileo does not differ much. At the same time, RMS of the phase measurements for SBAS signals is 1.5–3 times higher,

which is in accordance with the known results by [23]. The lowest noise level among the considered signal frequencies and components was detected for L5 frequency.

**Table 2.** RMS of the phase measurement noises of GNSS signal phase.

| Navigation System | Signal Component/RMS of the Noise (Cycles) | | | | |
|---|---|---|---|---|---|
| GPS | L1C 0.017 | L1W 0.017 | L2W 0.010 | L2X 0.014 | L5X 0.007 |
| GLONASS | L1C 0.014 | L1P 0.015 | L2C 0.013 | L2P 0.014 | - |
| Galileo | L1X 0.019 | | - | | L5X 0.006 |
| SBAS | L1C 0.035 | | - | | L5I 0.027 |

## 3. Discussion of Results

Our results are presented in the histogram form. To construct each histogram we evaluated the number of fd values for each particular case: one fd value per one spectrum of the phase variations. Each spectra was obtained by applying the Fast Fourier Transformation (FFT) from the 1-h carrier phase record for the particular satellite and signal component with 50-Hz sampling. Thus, the general statistics of this study includes ~2900 particular spectra of the phase variations (for instance, two days × 24 h × (6–8 GPS satellites in view × 5 GPS signal components + (4–6) GLONASS satellites × 4 GLONASS signal components + (2–4) GALILEO satellites × 2 GALILEO signal components) + SBAS data).

### 3.1. Deviation Frequency Estimates for Different GNSS Signal Components

Figure 3 shows the histograms of distribution of the deviation frequency (fd) for signal components L1X and L5X of GALILEO satellites during 24 h. The deviation frequency value varied within 15–22 Hz during both days. The most probable deviation frequency in either day of two and for all signal components was 20 Hz. The character of distribution shown for L1X and L5X components under the same conditions has no fundamental difference. At the same time, the carrier phase noise for the L5X component is significantly lower than for the L1X component (Table 2). Moreover, the histograms constructed for the control day of 13 April and the geomagnetically disturbed day of 16 April differ essentially. Histograms for the quiet day are characterized by more gradual rise at their left part. In contrast, the most probable deviation frequency is pronounced more clearly at the histograms for the disturbed day.

Figures 4 and 5 present the similar histograms for the signal components of GLONASS satellites and GPS satellites, respectively.

The character of distributions at histograms constructed for all GPS and GLONASS signal components under the same conditions has no fundamental difference as well. Deviation frequency again varies within 15–22 Hz. Though for some components this range is narrower—within 16–22 Hz (Figure 4a,c,h and Figure 5b,d,f). When comparing histograms in Figure 3 with ones on Figures 4 and 5 there are no significant differences between the histogram forms. However, the characteristic deviation frequency of 20 Hz in Figures 4 and 5 has higher probability under geomagnetically disturbed conditions (0.43–0.55 for GPS, GLONASS and ~0.35 for GALILEO satellites). It can be noted the smoother increase of deviation frequency at the lower frequencies area during the quiet day of 13 April (Figure 4a,c,e,g and Figure 5a,c,e,g,i). It is also worth noting the rather uniform appearance of the deviation frequency distribution in the histograms for GPS with the only exception of the 20 Hz peak during geomagnetically disturbed day (Figure 4 right panels).

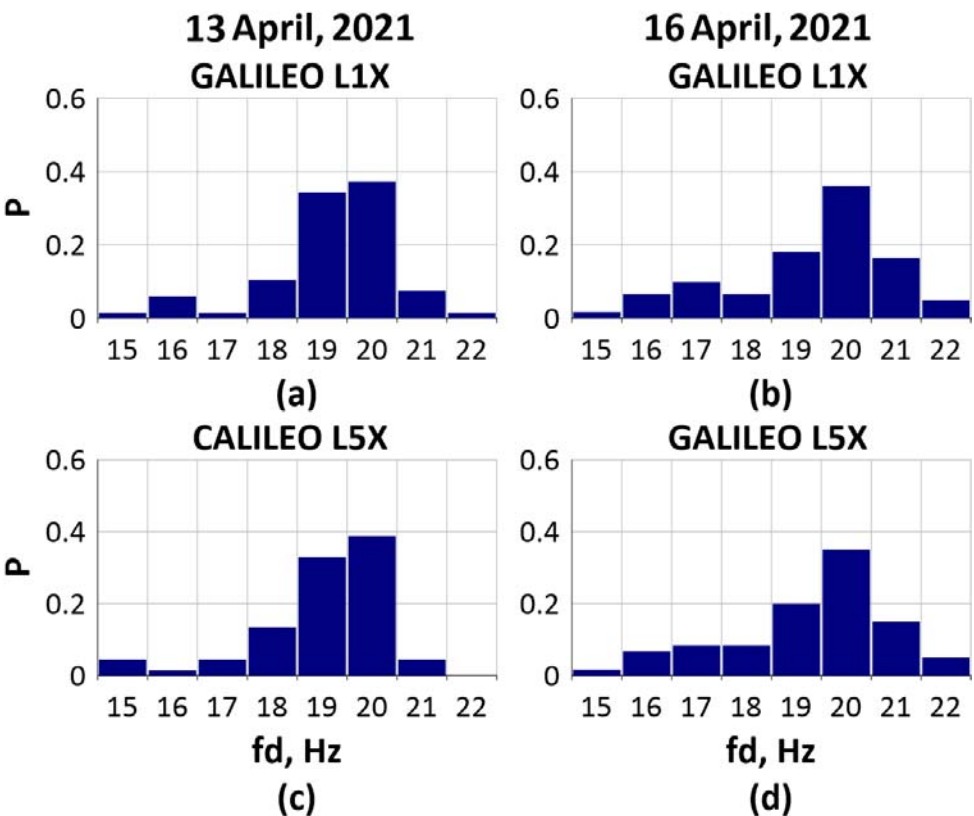

**Figure 3.** Deviation frequency distribution on 13 April 2021 (**a**,**c**) and 16 April 2021 (**b**,**d**) for GALILEO signals.

It is worth noting that regardless of the significant difference in the carrier phase noise (see Table 2) the results for GPS, GLONASS and GALILEO signals are quite similar. The difference in the phase noise did not affect the characteristic deviation frequency.

Figure 6 shows the distribution of deviation frequency of carrier phase for L1X and L5I components of SBAS satellite signals during 24 h. The measurements were obtained for three SBAS satellites whose angular characteristics are given in Table 3.

According to Table 3, all observed SBAS satellites were concentrated in one narrow southwest sector and at low elevation angles. Consequently, spatial variability of the ionosphere due to the line-of-sight movement through different ionospheric/atmospheric regions may not be taken into account.

The histogram form in Figure 6 differs significantly from the histograms shown in Figures 3–5. The SBAS deviation frequency varied within 13–20 Hz, which is smaller than one of GPS, GLONASS or Galileo. The characteristic deviation frequency can be determined with confidence only in one case of four in Figure 6 (in panel b). For example, the characteristic deviation frequency in Figure 6a is 18 Hz with probability of 0.25, and it is 15 or 16 Hz with the probability of 0.22. In contrast, in Figure 6b the characteristic deviation frequency is pronounced much clearer at 18 Hz. Other two histograms in Figure 6 show no clear maximum. On the quiet day of 13 April we recorded smaller deviation frequencies as compared to 16 April. Under disturbed conditions the maximum is more pronounced and varied within 18–19 Hz (Figure 6b,d).

### 3.2. Deviation Frequency Dependence on Geomagnetic Conditions

The previous section showed high stability of characteristic deviation frequency fd = 20 Hz under particular conditions in this case study. This allows us to suggest that this characteristic frequency is primarily associated with the boundary of the regular ionosphere under current conditions. To test this assumption we analyzed diurnal variations in the most probable deviation frequency.

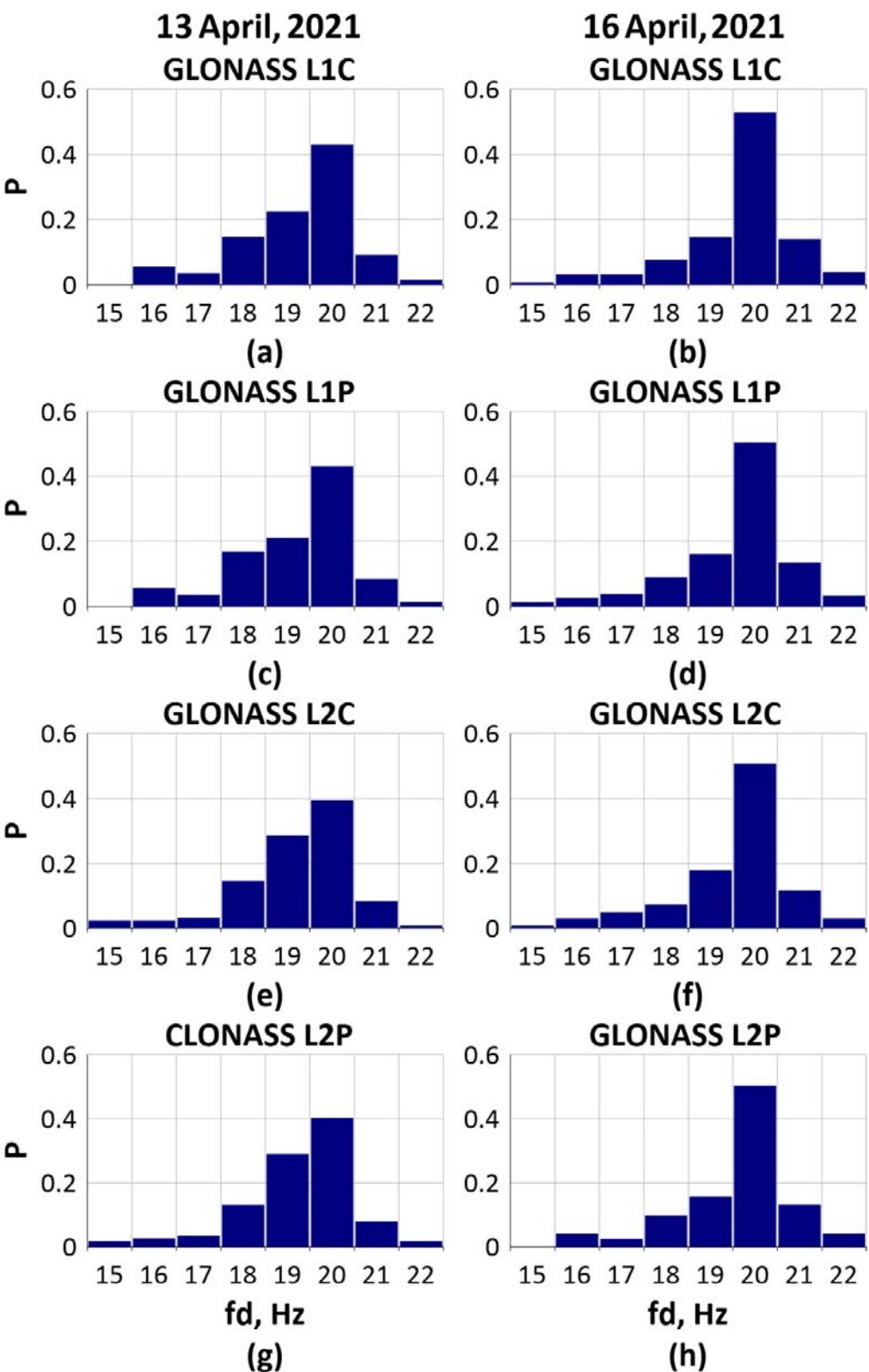

**Figure 4.** The same as in Figure 3 but for the GLONASS signal components (Table 1). Deviation frequency distribution on 13 April 2021 (**a,c,e,g**) and 16 April 2021 (**b,d,f,h**) for GLONASS signals.

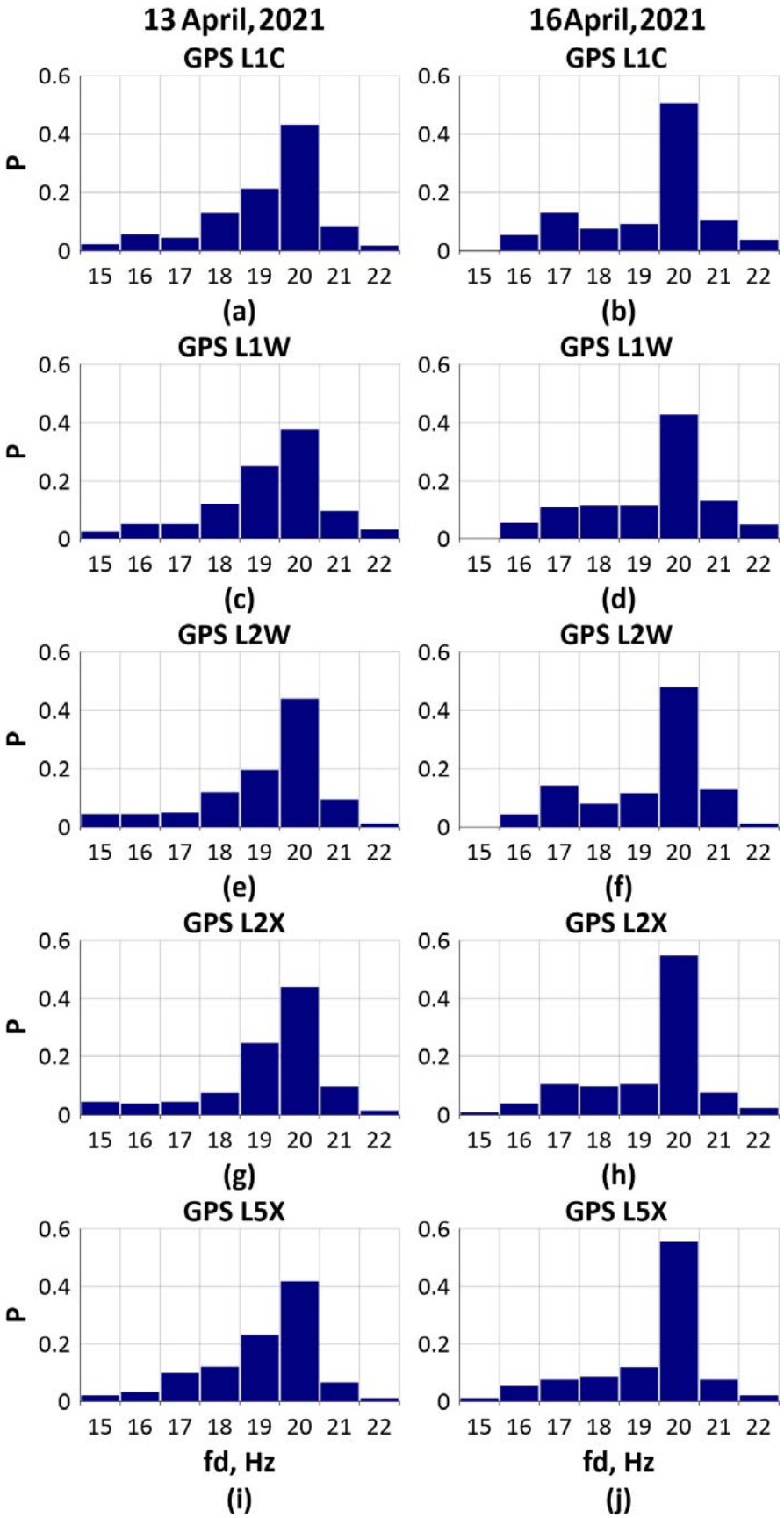

**Figure 5.** The same as in Figure 3 but for the signal components of GPS signals (Table 1). Deviation frequency distribution on 13 April 2021 (**a**,**c**,**e**,**g**,**i**) and 16 April 2021 (**b**,**d**,**f**,**h**,**j**) for GPS signals.

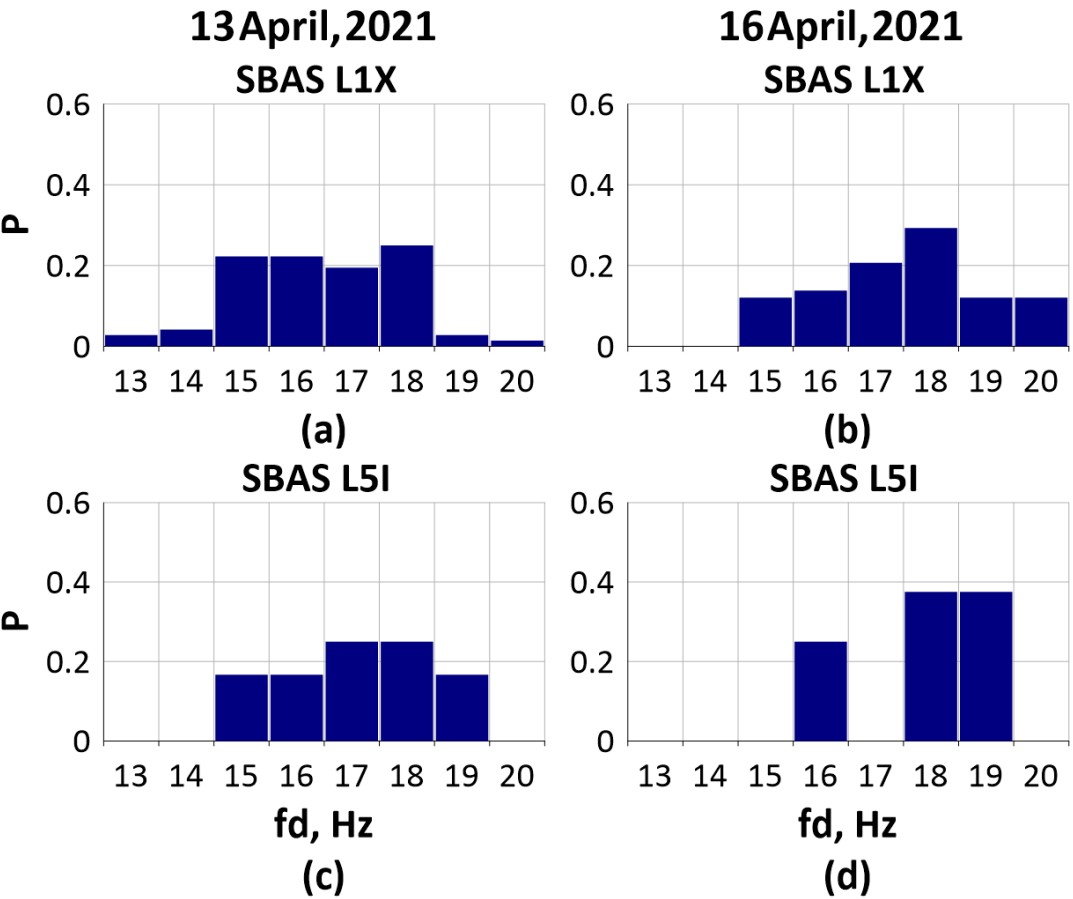

**Figure 6.** The same as in Figure 3 but for the L1X and L5I components of SBAS signals. Deviation frequency distribution on 13 April 2021 (**a**,**c**) and 16 April 2021 (**b**,**d**) for SBAS signals.

**Table 3.** Characteristics of SBAS satellite observations.

| SBAS Satellite Number | Mean Elevation, deg | Mean Azimuth, deg |
|---|---|---|
| S28 | 27.08 | 206.22 |
| S32 | 29.39 | 193.58 |
| S37 | 26.60 | 152.02 |

Figure 7 (upper panels) shows diurnal variations in the most probable deviation frequency for all signal components and for all navigation systems (excluding SBAS). We compute one fd value per one hour. The local noon and the night period indicated by the red arrow and blue rectangle, respectively. The left panels show the results for the quiet day, and the right panels show those for the disturbed day.

The lower panels of Figure 7 present the diurnal variations of the relative number of cases when *fd* = 20 Hz, calculated hourly as follows

$$N_{CFd} = \frac{N_{20}}{N_{TOT}} \tag{5}$$

where $N_{20}$ is the number of cases when *fd* = 20 Hz during one hour; $N_{TOT}$ is the total number of all fd values derived from all phase variations spectra for all satellites in view for all signal components and for all navigation systems (excluding SBAS) during one hour.

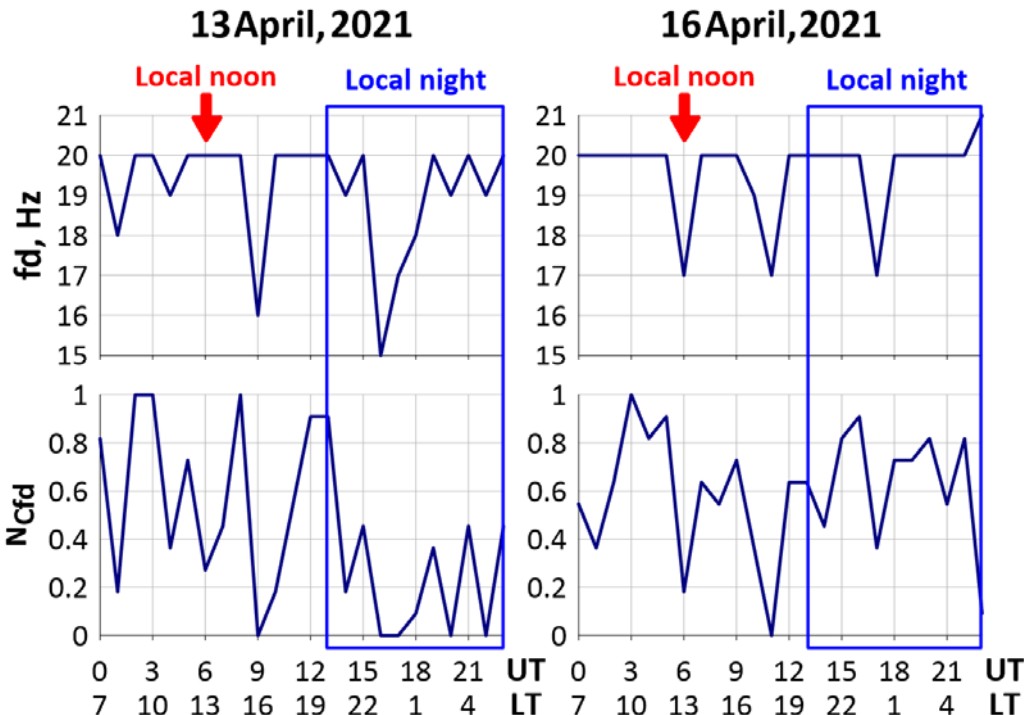

**Figure 7.** The deviation frequency variations (**upper** panels) and variations of the characteristic deviation frequency 20 Hz (**lower** panels) on 13 April (**left**) and 16 April (**right**).

Figure 7 (upper panels) proves that the deviation frequency at 20 Hz is typical during the most part of the day on 16 April (under weak geomagnetic disturbance). At the same time, there are short periods when the frequency drastically decreases to ~17 Hz. The first and the third decreases occurred near the time of local noon and local midnight. This differs on 13 April (the quiet day) when the deviation frequency varied more significantly (up to 15 Hz) and more frequently. The most profound drop of deviation frequency values (15 Hz) corresponded to the local midnight as in the previous case.

Figure 7 (lower panels) shows that the characteristic deviation frequency of 20 Hz was not always dominant during the 24 h. This means that the deviation frequency can take other values during most of the day. Unfolding of how the particular deviation frequency value is related to certain processes in the ionosphere is the issue for future research. In this case study no clear tendency in the character of $N_{CFd}$ variations was revealed. Nevertheless, the observed $N_{CFd}$ value at the particular moments was higher under quiet geomagnetic conditions ($N_{CFd} \leq 0.9$–1.0) than under disturbed conditions ($N_{CFd} \leq 0.8$).

## 4. Conclusions

We suggested the concept of the characteristic deviation frequency to determine the optimal sampling rate of the GNSS carrier phase data for the ionosphere studies. Current research considers that the characteristic deviation frequency is optimal if its further increase does not provide more information on the small-scale structure of the ionosphere.

Our case study showed that the characteristic deviation frequency tends to be stable during the short observation intervals (1 h). However, its value can vary significantly during 24 h (within 15–21 Hz). This agrees with the current knowledge on the physical ionospheric structure [24–28].

The maximal probability of the characteristic deviation frequency of 20 Hz at some moments did not exceed 0.8 under geomagnetically disturbed conditions and reached 0.9–1.0 under quiet conditions. The increase in this probability can indicate the increase in the spatial-temporal stability of the ionosphere under quiet conditions. Correspondingly, its decrease implies the presence of the disturbance effects that change the phase spectrum slope. This allows us to assume that the deviation frequency can be considered as a

promising index to estimate the boundary of non-variability of the ionosphere at the presence or absence of large-scale ionospheric disturbances.

The probability distribution of the characteristic deviation frequency obtained from SBAS data differs from the probability distribution of the other GNSS signals significantly. We showed that the carrier phase noise for SBAS signals is notably higher than for the signals of other GNSS. This is in accordance with the results of [23]. It is probable that the plasmasphere could impact the characteristic deviation frequency for SBAS signals. This hypothesis should be tested with further experiments and analysis of data. Moreover, the signals of geostationary SBAS satellites passed from the same narrow angular sector. This means that spatial variability of the ionosphere due to the line-of-site movement through different ionospheric/atmospheric regions did not impact on the deviation frequency variations significantly. This circumstance is absent for the radio propagation of signals from medium-orbit (and lower orbital) GNSS constellations. The data of geostationary BeiDou satellites which is characterized by the same phase noise level as GPS/GLONASS may be used for checking this hypothesis further.

**Author Contributions:** V.D. developed the conceptualization of this work; V.D. and Y.Y. designed the experiments; E.D. and M.S. provided the data processing, and performed the simulations and the experiments; all the authors participated in the analysis of the results; V.D. and M.S. prepared the manuscript with contributions from all the authors. All authors have read and agreed to the published version of the manuscript.

**Funding:** This work was performed under the Russian Science Foundation Grant No. 17-77-20005.

**Institutional Review Board Statement:** Not applicable.

**Informed Consent Statement:** Not applicable.

**Data Availability Statement:** SYM-H data were obtained from the NASA/GSFC's Space Physics Data Facility's OMNIWeb service (https://omniweb.gsfc.nasa.gov. Last access: 24 November 2021). Kp-index data were obtained from GFZ German Research Centre for Geosciences (ftp://ftp.gfz-potsdam.de. Last access: 24 November 2021) [29].

**Acknowledgments:** We thank Artem Vesnin for his help in GNSS data acquisition. The study used equipment of the Center for Common Use «Angara» of ISTP SB RAS (http://ckp-rf.ru/ckp/3056/. Last access: 24 November 2021) operating under Ministry of Science and Higher Education of the Russian Federation.

**Conflicts of Interest:** The authors declare no conflict of interest.

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
