# Peer review of "Experimental Estimation of Deviation Frequency within the Spectrum of Scintillations of the Carrier Phase of GNSS Signals"

_remotesensing, doi:10.3390/rs13245017_

Round 1

Reviewer 1 Report

The authors attempt to characterize the deviation frequency of the phase scintillation spectrum.   Although the subject is interesting and deserves investigation, my main concern is that the amount of data being presented (only two days and a single receiver) does not allow to draw clear conclusions.  Are the results applicable in a general sense, or are they specific to the local ionospheric conditions during those days and to the PLL parameters of that particular receiver?  The relevance of the paper would increase if more data would be processed, including different receiver types.

Some points to be addressed:

Figure 1: the figure shows the Bz component but it is never mentioned in the text.  For clarity, either Bz should be discussed in the text, or removed from the figure.

Line 80: what does “sudden commencement feature” mean and how is it related to the study?

Lines 101-107: it would help a lot to illustrate this paragraph with a real spectrum where the “break point” and the slope are shown, so that readers clearly understand what fd corresponds to.

Table 2: how to interpret the unit “2pi cycles”?    Does “1” correspond to 1 cycle, i.e. 2pi radians, or to 2pi cycles, ie. 4pi^2 radians?  I assume the unit is “cycles” and not “2pi cycles”.

Section 3.1: to help interpreting the histograms, please explain how many fd samples are evaluated per day.  Is it one fd per hour, or per minute?

Figure 6: I couldn’t understand what the lower panel represents.  The text says “the variation of the characteristic deviation frequency 20Hz during 24hours”, but then why is it shown per hour? How is this “variation” computed.   This should be clarified in the text.

Line 201-202: “also characterize the state of the regular transionospheric radio channel”.  This aspect has not been addressed in the paper, and it should therefore be removed from the conclusion.

Some typos:

Figure 2: Abril àApril

Figure 6: devided à divided.

Author Response

We thank the Editor and the Reviewers for their valuable comments.

The manuscript was improved according to your suggestions.

Rewiever#1

The authors attempt to characterize the deviation frequency of the phase scintillation spectrum.   Although the subject is interesting and deserves investigation, my main concern is that the amount of data being presented (only two days and a single receiver) does not allow to draw clear conclusions.  Are the results applicable in a general sense, or are they specific to the local ionospheric conditions during those days and to the PLL parameters of that particular receiver? 

In general, we agree with this remark and the results we obtained need for further extended explorations. We studied the characteristic deviation frequency under particular observation conditions but for different constellations (GPS, GLONASS, GALILEO and SBAS). ~2900 spectra of phase variations (based on high-rate data) were analyzed.  The goal was to test the idea of using the deviation frequency for the ionospheric studies, not to provide the statistical results. To take this remark into account we added some new explanations in the text (Please look in lines 156-163. We also point about the case study in the text lines 20, 254, 262 in the revised manuscript).

The relevance of the paper would increase if more data would be processed, including different receiver types.

We completely agree that more data from different GNSS receiver types would greatly improved our paper. At the same time, we would like to emphasize the difficulty of acquiring the high resolution data obtained at the same observatory using different precise GNSS receivers. The ideal case would be the measurements performed at the same time, place and with the same common antenna of course. But it is limited with our real opportunities. We had to involve the available data.

Some points to be addressed:

Figure 1: the figure shows the Bz component but it is never mentioned in the text.  For clarity, either Bz should be discussed in the text, or removed from the figure.

The information on Bz was removed as it was redundant. Figure 1 was changed.

Line 80: what does “sudden commencement feature” mean and how is it related to the study?

The sentence was withdrawn as it was unnecessary.

Lines 101-107: it would help a lot to illustrate this paragraph with a real spectrum where the “break point” and the slope are shown, so that readers clearly understand what fd corresponds to.

Thank you for your comment. We added Figure 2 and the correspondent text (lines 133-135 and 141-145) in the new version of the manuscript.

Table 2: how to interpret the unit “2pi cycles”?    Does “1” correspond to 1 cycle, i.e. 2pi radians, or to 2pi cycles, ie. 4pi^2 radians?  I assume the unit is “cycles” and not “2pi cycles”.

Thank you for your comment. The unit was changed as “cycles” (i.e. 1 cycle=2pi rad).

Section 3.1: to help interpreting the histograms, please explain how many fd samples are evaluated per day.  Is it one fd per hour, or per minute?

We evaluate one fd value per one spectra of phase variations: one spectra was obtained from 1-hour carrier phase record with 50-Hz sampling. Thus the general statistics of this study includes ~ 2900 particular spectra of the phase variations: for instance, two days*24 hours * (6-8 GPS satellites in view * 5 GPS signal components + (4-6) GLONASS satellites *4 GLONASS signal components + (2-4) GALILEO satellites *2 GALILEO signal components) + SBAS data). To take this remark into account we added this explanation in the revised text (please see Lines156-163).

Figure 6: I couldn’t understand what the lower panel represents.  The text says “the variation of the characteristic deviation frequency 20Hz during 24hours”, but then why is it shown per hour? How is this “variation” computed.   This should be clarified in the text.

You are right. Thank you. The figure was corrected (Please see Figure 7 in the new version of the manuscript) and the corresponding explanations were added to the text (Lines 232-239 and 250-256).

Line 201-202: “also characterize the state of the regular transionospheric radio channel”.  This aspect has not been addressed in the paper, and it should therefore be removed from the conclusion.

We agree. The sentence was withdrawn from the revised manuscript.

Some typos:

Figure 2: Abril àApril

The typo was corrected. Thank you.

Figure 6: devided à divided.

The figure (Fig 7 in the revised manuscript) was corrected. Thank you.

Reviewer 2 Report

The work ”Experimental estimation of deviation frequency within the spectrum of scintillations of the carrier phase of GNSS signals” presents a simple case study aimed at comparison of frequency deviation in different geomagnetic conditions. In opinion of the reviewer the work does not provide details of algorithm and, consequently, is difficult to follow. The major concerns are:

  1. The introduction is too brief and should provide more details on the presented topic.
  2. The analysis, including two periods characterized by different geomagnetic/ionospheric activity, should present a comparison of raw (or slightly processed) data to show a difference for both datasets. Instead of this only final results of fd are given.
  3. The entire procedure (lines 95-98) is definitely too short and thus, completely unclear. Please provide more details, because without this step the results cannot be understood.
  4. The example FFT results with a “break point” should be added. What is an accuracy of fd computation?
  5. “The second-order derivative of the signal carrier phase was used as the noise magnitude…” Is it an appropriate approach for high-rate data which may be affected by a temporal correlation? How the RMS was converted to undifferenced data?
  6. I do not see any connection between first results (Table 2) and the discussion given in point 3.
  7. In the opinion of reviewer the results in figures 2-4 seem to be consistent. It suggests there is no impact of weak geomagnetic activity on fd, which is usually close to 20 Hz. It is not clear, however, what this value gives for ionospheric studies.
  8. I would not say that the lower panels of figure 6 depict a highly variable fd. According to the top panels only for 3-4 hours fd is not equal 19-20 Hz. The results are probably different because you used only data with fd=20 Hz.
  9. “The probability of observation of the characteristic deviation frequency 20Hz at some particular time moments was higher under quiet conditions on April 13th (up to 0.9-1.0) than under disturbed conditions (up to 0.8) during 24 hours.” Please note that for the other hours the probability was higher for disturbed period. Please avoid such definite statements for variable results.

Author Response

We thank the Editor and the Reviewers for their valuable comments.

The manuscript was improved according to your suggestions.

Reviewer#2

Comments and Suggestions for Authors

The work ”Experimental estimation of deviation frequency within the spectrum of scintillations of the carrier phase of GNSS signals” presents a simple case study aimed at comparison of frequency deviation in different geomagnetic conditions. In opinion of the reviewer the work does not provide details of algorithm and, consequently, is difficult to follow. The major concerns are:

  1. The introduction is too brief and should provide more details on the presented topic.

Thank you for your comment. Our work is based on the results and definitions by (McCaffrey and Jayachandran, 2017) and our case study is very specific. We think that it would be irrelevant to repeat the information from this paper again. Also we explain briefly the concept in lines 61-69 in the revised manuscript.

  1. The analysis, including two periods characterized by different geomagnetic/ionospheric activity, should present a comparison of raw (or slightly processed) data to show a difference for both datasets. Instead of this only final results of fd are given.

We agree that this is a fair point in general. At the same time, in our case the “slightly processed raw” data are the de-trended phase records of high temporal resolution (50 Hz sampling). It is hardly possible to demonstrate the visual difference between data measured under different ionospheric activity conditions (especially when scintillations can occur). This is why, we added Figure 2 that illustrates the identification of the deviation frequency under quiet and geomagneticaly disturbed conditions. We hope this will do.

  1. The entire procedure (lines 95-98) is definitely too short and thus, completely unclear. Please provide more details, because without this step the results cannot be understood.

Thank you for your comment. We added the description of data pre-processing procedures (Please see Equations 1-4 and Lines 101-127 in the revised manuscript).

  1. The example FFT results with a “break point” should be added. What is an accuracy of fd computation?

Thank you for your comment. We added an example of FFT with a “break point” (Please see Figure 2 in the revised manuscript). The deviation frequency was obtained visually from each particular spectrum with accuracy of ~0.5 Hz.

  1. “The second-order derivative of the signal carrier phase was used as the noise magnitude…” Is it an appropriate approach for high-rate data which may be affected by a temporal correlation? How the RMS was converted to undifferenced data?

The most convenient approach to extract the carrier phase noise from the complex phase ranging data is to use phase derivatives. It allows us to receive the phase noise directly from the phase measurements without additional processing procedures. One can estimate an instantaneous signal phase value at the phase lock loop (PLL) filter output (at each i-th instant time) based on the discrete Markov’s chain model:

where TCOR is the PLL pre-detection integration time and  is the forming zero-mean Gaussian noise of the phase. First and second components describe the phase ranging trend and its slow changing. The last component in Equation is the second-order derivative of the phase corresponds to the phase noise.

Based on this we computed RMS of the phase noise from the second order derivative of the phase records with 50 Hz sampling. The question about the temporal correlation for high-rate data is mostly about the lowest sampling rate which defines the border between the noise and the correlated phase variations. According to [GNSS High-Rate Data and the Efficiency of Ionospheric Scintillation Indices. DOI: 10.5772/intechopen.90078] the sampling rate ~10 Hz is this border. It means that the second order derivation of the phase mostly brings the phase noise in case if the data sampling rate is >10 Hz. Sometimes it is possible that high rate data contain not only noise but also correlated high frequency ionospheric phase scintillations and satellite oscillator anomalies mimicking ionospheric phase scintillation [GPS Solut (2014) 18:387–391. DOI: 10.1007/s10291-013-0338-4]. Such events are rather rare and we did not take it into account in RMS computation.

  1. I do not see any connection between first results (Table 2) and the discussion given in point 3.

It is a fair point, thank you. To take this remark into account we added correspondent explanations into the text (Please see Lines 141-145, 191-194, 274-275).

  1. In the opinion of reviewer the results in figures 2-4 seem to be consistent. It suggests there is no impact of weak geomagnetic activity on fd, which is usually close to 20 Hz. It is not clear, however, what this value gives for ionospheric studies.

Thank you for your comment. Some considerations were added in lines 61-69 and 270-272.

  1. I would not say that the lower panels of figure 6 depict a highly variable fd. According to the top panels only for 3-4 hours fd is not equal 19-20 Hz. The results are probably different because you used only data with fd=20 Hz.

You are right. The figure (now Figure 7) and its description were corrected (Please see Lines 232-239).

  1.  “The probability of observation of the characteristic deviation frequency 20Hz at some particular time moments was higher under quiet conditions on April 13th (up to 0.9-1.0) than under disturbed conditions (up to 0.8) during 24 hours.” Please note that for the other hours the probability was higher for disturbed period. Please avoid such definite statements for variable results.

Thank you for this comment. We corrected the text according to this remark (please see Lines 250-256 in the revised manuscript).

Round 2

Reviewer 2 Report

The authors have provided satisfactory answers to all my points.